# TWO ARE BETTER THAN ONE: UNCERTAINTY-AWARE VISION-LANGUAGE MODELS FOR VIDEO ANOMALY DETECTION

## ABSTRACT

Vision-language models (VLMs) have demonstrated impressive reasoning capability in visual understanding tasks. One recent highlight of VLMs is their success in generating human-understandable explanations in video anomaly detection (VAD), which is an advanced video understanding task requiring delicate judgment on context-dependent and ambiguous video content. Representative works mainly formulate this problem as a natural language generation task conditioned on task-related prompts and visual inputs. However, under this paradigm, the input is processed segment by segment, and VLMs generate a response for each segment independently, which inevitably leads to uncertainty in their reasoning with a limited context. To bridge this fundamental gap, we propose an uncertainty-aware VLM framework named UNA for VAD to objectively identify the reasoning-level uncertainty in VLMs and correspondingly mitigate it: Firstly, UNA obtains relevant scenes by temporal and semantic relevance and determines the existence of uncertainty by the prediction consistency across relevant scenes. After that, collective intelligence via the cooperation of VLMs is introduced to address the uncertainty. With UNA, VLMs can achieve remarkable performance and advanced explainability, surpassing task-specific methods in challenging benchmarks in the most difficult setting where instruction tuning is not allowed for the first time.[1]

## 1 INTRODUCTION

Video anomaly detection (VAD) has received significant interest within the AI research community due to its tremendous benefit in achieving automated decision-making for safety-critical applications, including video surveillance (Ramachandra et al., 2020), autonomous driving (Yao et al., 2022), and medical diagnosis (Fernando et al., 2021). An ideal VAD system is expected to output prediction results that are (1) **accurate** — correctly localizing the

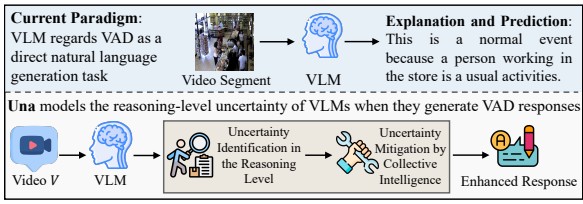

Figure 1: Existing methods directly model VAD as a natural language generation task but ignore the uncertainty in their reasoning. We propose UNA to solve that.

occurrence of video anomalies, (2) **explainable** — generating easily interpretable VAD explanations for human users, and (3) **generalizable** — consistently performing VAD as intended in any given scenario. To achieve these goals, the research community has continued to advance deep neural network (DNN) architectures for VAD. Particularly, vision-language model (VLM) (Hurst et al., 2024) is a game-changer in pursuing ideal VAD performance.

Integrating vision transformers (Dosovitskiy et al., 2021) as "eyes" and large language models (LLMs) as "brains", VLMs have superior ability to perceive visual inputs, follow textual instructions, and generate verbalized prediction results. Existing research has explored applying VLMs to VAD by formulating it as a natural language generation task (shown in Fig. 1), using either prompt engineering (Ye et al., 2025) to guide reasoning or instruction tuning (Zhang et al., 2024a) to adapt

---

[1]The implementation code is available in the supplementary material and will be made publicly available.

VLMs to specific task prompts. During inference, video segments are extracted from a video and processed by a single VLM to detect anomalies, and the VLM provides language-based explanations.

**Key Observations and Research Question.** Despite the advantages of the aforementioned pipeline, it overestimates the predictive power of VLMs in VAD. Since anomalies are diverse and the context of a video segment is short, VLMs are not perfectly certain regarding the responses they generate. Thus, a more realistic viewpoint for designing VLMs for VAD should be adopted: VLMs are not omniscient, and they are not always certain about their predictions. That is, existing VLM-based VAD approaches overlook the uncertainty in the reasoning of VLMs during inference. In this paper, we aim to bridge this non-negligible pitfall by investigating the following important question:

*How can we model the uncertainty in the reasoning of VLMs and mitigate it to output reliably accurate and explainable predictions?*

**Our Approach.** This research question is fundamental to achieving the ideals of VAD. Specifically, by introducing a novel uncertainty-aware mechanism into VLMs, we can identify scenes where the model exhibits uncertainty. With a mitigation process in place, VLMs can correct potential errors in these uncertain segments, preventing their propagation into final decisions. This not only enhances VAD performance but also enables error correction for initial explanations. As a result, decisions made by uncertainty-aware VLMs can make VLMs generalizable in different scenarios.

Motivated by this, in this paper we investigate the design of an uncertainty-aware video anomaly detection framework with VLMs with treatments of uncertainty identification and mitigation, termed as Una. Firstly, Una will determine whether the VLM is uncertain on a given video segment by its prediction consistency with its relevant segments. Based on the properties of video inputs, Una obtains relevant scenes by temporal and semantic relevance. After that, Una resolves the uncertainty caused by individual reasoning via collective intelligence with VLMs forming as a team. By uncertainty identification and mitigation in Una, the deployed VLMs in VAD can adaptively re-evaluate its decision on the existence of anomalies in videos when facing uncertainty.

**Contributions**. In this work, we make the following key contributions:

* To our knowledge, we are the first to champion an uncertainty-aware perspective for utilizing VLMs for VAD. This new perspective reflects real-world scenarios, where VLMs unavoidably face reasoning-level uncertainty because the provided limited context makes it difficult for them to reach fully certain decisions in the presence of ambiguous and diverse video anomalies. This conceptual shift represents a significant and timely contribution to the field of VAD.

* Under this concept, we design a novel VAD framework named Una equipped with uncertainty identification and mitigation as the first attempt. Specifically, Una includes a new uncertainty identification principle by the prediction consistency of VLMs among relevant segments, which is customized for VAD and can be broadly applied in video understanding tasks. Meanwhile, Una pioneers the technique of collective intelligence in mitigating reasoning-level uncertainty in VAD.

* Una can empower VLMs with an advanced reasoning ability that corrects misjudgment caused by reasoning-level uncertainty. Extensive experimental results demonstrate that the proposed Una renders VLMs to improve their VAD ability in challenging VAD benchmarks including UCF-Crime and XD-Violence with outstanding numerical performance and persuasive explanations.

## 2 RELATED WORK

**Video Anomaly Detection.** Detecting video anomalies is inherently challenging due to the ambiguous and context-dependent nature of what constitutes an anomaly. Previous to VLMs, commonly used are task-specific DNN structures trained with weakly supervised classification tasks (Sultani et al., 2018; Wu et al., 2024b) or unsupervised frame reconstruction tasks (Liu et al., 2018; Ye et al., 2019; Lu et al., 2013). However, they can output numerical results only and largely overlook the explainability aspect. To address this limitation, the recent line of literature introduces VLMs to generate predictions and explanations together. Prompt engineering (Zanella et al., 2024; Ye et al., 2025) or instruction tuning (Zhang et al., 2024a; Lv & Sun, 2024; Yang et al., 2024a; Tang et al., 2024) is adopted to make VLM responsive to the instruction for VAD. However, VLMs will have uncertain reasoning in this language generation process for the visual context is limited during the generation process. In light of this, we build an uncertainty-aware framework named Una to address this issue.

**Uncertainty-Aware LLMs.** Uncertainty-aware mechanisms have been found useful in LLM reasoning. The intuition is that by making the chain-of-thought (Wei et al., 2022) in LLMs uncertainty-aware, LLMs can focus their reasoning on the uncertain part and generate more satisfactory responses. Existing works have investigated the uncertainty-aware design in token generation tasks like code generation (Zhu et al., 2025) and reasoning tasks like medical diagnosis (Hu et al., 2024), troubleshooting (Hu et al., 2024), and question answering (Nikitin et al., 2024; Ye et al., 2024). However, uncertainty in reasoning remains largely underexplored in VAD despite its importance. Our work bridges that by introducing a novel framework to model uncertainty in the reasoning of VLMs.

**Collective Intelligence in LLMs.** Collective intelligence refers to the phenomenon where groups of individuals, whether human or artificial agents, achieve higher problem-solving capacity than any individual working alone (Malone et al., 2010). This principle has inspired recent developments in AI including multi-agent collaboration (Zhang et al., 2024b) such as AutoGen (Wu et al., 2024c) and CAMEL (Li et al., 2023), which has been actively explored as an effective means to enhance reasoning and decision-making. In this work, we explore building collective intelligence through the cooperation between VLM agents to tackle uncertain scenarios in VAD.

## 3 THE UNA FRAMEWORK

We now detail our motivation by discussing how VLM reasons in VAD and later the design of UNA.

### 3.1 MOTIVATION: EXISTING VLM-BASED INFERENCE IN VAD HAS LIMITATIONS

**Input and Output.** During inference in VAD, a video $V = \{I_i\}_{i=1}^{F}$ with $F$ frames is given, where $I_i$ is the $i$-th frame. Frame-level ground truth $Y = [y_1, \ldots, y_F]$ is provided: $y_i = 1$ if $I_i$ contains an anomaly, and $y_i = 0$ otherwise. The employed VLM $f$ (base model) is expected to generate an anomaly score sequence $\hat{Y} = [\hat{y}_1, \cdots, \hat{y}_F]$ for all frames, where each $\hat{y}_i \in [0, 1](1 \le i \le F)$.

**Video Anomaly Scoring by VLMs.** However, it is impractical for $f$ to compute the anomaly score for each frame individually for $F$ is huge. Thus, $f$ follows a coarse-to-fine procedure shown in Fig. 2. It generates segment-level scores in Stage 1 and refines them in Stage 2. To detail:

*Stage 1: Initial Anomaly Scoring by Natural Language Generation.* First, a given video $V$ will be divided into $h$ segments $\{v_j\}_{j=1}^{h}$. Each segment $\{v_j\}$ contains $\kappa$ frames uniformly sampled from a window, e.g., a 10-second glimpse. VAD is formulated as a natural language generation process for each segment $v_j$: a textual VAD-specific instruction $\theta$ is input to $f$ and prompt $f$ to generate an explanation $E_j$ whether an anomaly exists. If $E_j$ contains analysis that an anomaly exists in $v_j$, $f$ returns a numerical output $\tilde{y}_j = 1$, and $\tilde{y}_j = 0$ otherwise.

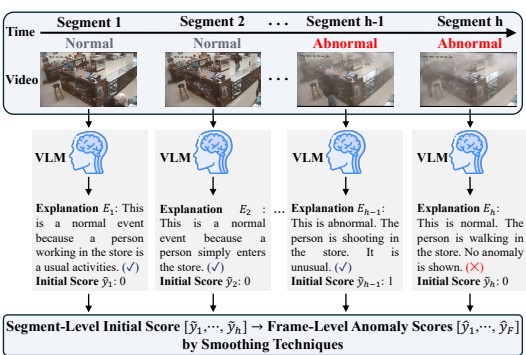

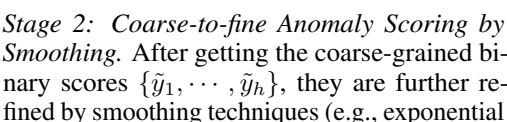

*Stage 2: Coarse-to-fine Anomaly Scoring by Smoothing.* After getting the coarse-grained binary scores $\{\tilde{y}_1, \cdots, \tilde{y}_h\}$, they are further refined by smoothing techniques (e.g., exponential

Figure 2: Existing VLM-based pipelines conduct anomaly scoring for VAD with a coarse-to-fine process. Uncertainty exists due to the limited context.

moving average (Yang et al., 2024a) and Gaussian smoothing (Ye et al., 2019)) to obtain a set of continuous anomaly scores for all frames as $\hat{Y} = [\hat{y}_1, \ldots, \hat{y}_F]$, which is expected to be closer to $Y$.

**Limitations of Existing VLM-Based Pipelines for VAD.** In VAD, VLMs map each video segment independently to explanations and numerical scores in the language space via a natural language generation formulation. Consequently, constrained to isolated segments, the lack of broader context and the limited evidence makes the reasoning inherently contain uncertainty. We term this type of uncertainty as **reasoning-level uncertainty**. The uncertain responses will unavoidably harm the accuracy and explainability of VLMs in VAD: Firstly, the mistakes they cause during the initial scoring phase can propagate into the fine-grained frame-level scores during the coarse-to-fine scoring, which eventually degrades the VAD performance. Secondly, the explanation of an uncertain response

can misinterpret the observed scenes and output unsatisfactory explanation for humans. Thus, it is imperative to study this type of uncertainty for VLMs in VAD by investigating two fundamental questions: (1) how to identify such uncertainty? and (2) how to mitigate it when it is identified? In this paper, we answer these questions by proposing an uncertainty-aware framework, UNA. Details are as follows and the full pseudocodes are in Alg. 1 in Sec. A.1 in Appendix.

### 3.2 IDENTIFYING REASON-LEVEL UNCERTAINTY VIA RELEVANT SCENES

To identify the uncertainty during reasoning, UNA first includes a test determining whether $f$ is uncertain in its generated response $E_j$ (codified by $\tilde{y}_j$) for each $v_j$. Previous works provide us with two straightforward solutions to that by relying on either the internal token probabilities (Zhu et al., 2025; Hu et al., 2024) produced during generation or the self-consistency (Lin et al., 2024) of the repeatedly generated responses:

**(Option 1) Token Probability-Based Uncertainty Test**: *Given a video segment $v_j$ and instruction $\theta$, the VLM $f$ generates a decision token ('0' for normal or '1' for anomaly) during the natural language generation process. Along with the token, $f$ also produces the associated probabilities $p_0$ and $p_1$ for the two possible outcomes. The uncertainty can be defined by the information entropy (Shannon, 1948) $H(v_j) = -(p_0 \log p_0 + p_1 \log p_1)$ of this binary distribution. Uncertainty exists if the entropy is larger than a threshold $\delta$, as expressed by the indicator function $U_{token}(v_j) = \mathbb{1}\{H(v_j) > \delta\}$.*

**(Option 2) Self-Consistency-Based Uncertainty Test**: *Given a video segment $v_j$ and instruction $\theta$, we query the VLM $f$ repeatedly for $N$ trials, obtaining a set of responses $\{f^{(1)}(v_j), f^{(2)}(v_j), \ldots, f^{(N)}(v_j)\}$. If all responses are identical, the VLM is considered confident on $v_j$. Otherwise, the disagreement among responses indicates uncertainty. Formally, the uncertainty can be expressed using an indicator function: $U_{\text{self}}(v_j) = \mathbb{1}\{\exists i, l \text{ such that } f^{(i)}(v_j) \neq f^{(l)}(v_j)\}$, where $U_{self}(v_j) = 0$ denotes no uncertainty and $U_{self}(v_j) = 1$ denotes uncertainty.*

***Discussion: Limitations of Existing Solutions***. Despite their popularity in LLM pipelines, these options are either ineffective or impractical in modeling the reasoning-level uncertainty of VLM in VAD. Firstly, Option 1 primarily serves for token generation tasks such as code generation and question answering. It is inapplicable because such uncertainty applied in VAD only reflects the uncertainty of claiming the existence of anomalies with the previously generated words in language decoding. Secondly, for Option 2, repeatedly computing the generated response multiple times is impractical for video data. Due to the huge amount of video segments, the computation increases to $N$ times of the original one, which makes the overhead too demanding. We verify this argument in experiments (see Table 3) and find them inferior in discovering uncertainty. Thus, we propose a novel measurement tailored for VAD that objectively reflects the reasoning-level uncertainty:

> **Reasoning-Level Uncertainty Test via Relevant Scenes**: *Given a target segment $v_j$, we first retrieve $n$ relevant scenes $\{r_1(v_j), r_2(v_j), \ldots, r_n(v_j)\}$ based on certain relevance measurements. We have anomaly scores for both the target and its relevant scenes from the VLM. If the predictions across this set are consistent, the VLM is considered confident in $v_j$; otherwise, disagreement indicates reasoning-level uncertainty. Formally, the uncertainty is defined as:*
>
> $$U_{\text{relevant}}(v_j) = \mathbb{1}\{\exists i \text{ such that } f(v_j) \neq f(r_i(v_j))\}, \tag{1}$$
>
> *where $U_{relevant}(v_j) = 0$ denotes consistent reasoning across relevant scenes, while $U_{relevant}(v_j) = 1$ flags reasoning-level uncertainty.*

***Our Insight***. The proposed test for reasoning-level uncertainty will be both *effective* and *practical*. Firstly, by relevant scenes, we mean those containing similar semantic meaning to a given $v_j$. If two inputs $v_i$ and $v_j$ encode semantically similar content, their predictions from $f$ should agree. Otherwise, the reasoning of $f$ is unstable with respect to semantic equivalence, which represents the existence of reasoning-level uncertainty. This is analogous to human reasoning (Tversky & Kahneman, 1981): *when people give different answers to rephrased but semantically equivalent questions, it is a sign of weak belief strength from the psychology perspective*. Secondly, such computation does not introduce extra overhead because every segment will be input to $f$ only once. It can be easily implemented by reusing previously computed results. Thus, utilizing inconsistency across semantically similar inputs is a good behavioral proxy for reasoning-level uncertainty.

***Implementation.*** Under this idea, the key computation in implementation would be obtaining semantically similar scenes, i.e., evidence scenes that the reasoning of VLM should agree. Two properties of video data stand out when we consider the definition of relevant scenes: (1) Videos are temporal data where neighboring scenes usually represent gradual changes in the same underlying event. The semantic meaning does not change much. (2) Videos contain huge redundancy with repetitions or variations of the same semantic patterns occurring across time. Based on these properties, as shown in Fig. 3 with an example video, we

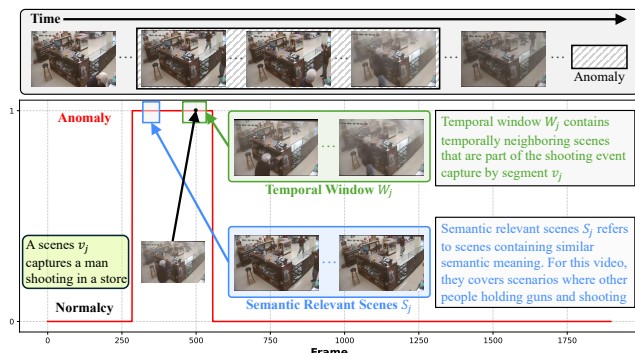

Figure 3: UNA uses the prediction consistency among relevant scenes decided by temporal and semantic information to decide if reasoning-level uncertainty exists.

would like to utilize *temporal relevance* and *semantic relevance* to obtain relevant scenes. In a confined setting of the input $V$, a VLM should reason consistently across these semantically relevant scenes if its knowledge can be well applied. Thus, we implement the test in Eq. (1) as follows:

Determining Reasoning-Level Uncertainty via Temporal Relevance. Since anomalies do not occur in a single isolated frame but span a local temporal window, we focus on the temporal window $W_j = [v_{j-\frac{w-1}{2}}, \cdots, v_{j-1}, v_j, v_{j+1}, \cdots, v_{j+\frac{w-1}{2}}]$ which includes $w-1$ neighboring scenes centered around $v_j$, where $w$ is the window size ($w$ is set small to ensure the semantic does not change). $f$ is deemed certain on $v_j$ if it outputs the same score for all segments. We denote it by:

$$U_j^{(\text{tem})} = \mathbb{1}\{\exists\, i, l \in \text{ such that } \tilde{y}_i \neq \tilde{y}_l\}, \tag{2}$$

where $i, l \in \{j - \frac{w-1}{2}, \cdots, j + \frac{w-1}{2}\}$, and $\tilde{y}_i$ and $\tilde{y}_j$ are the initial score given by the VLM. $U_j^{(\text{tem})} = 1$ indicates the existence of reasoning-level uncertainty in VLM, and $U_j^{(\text{tem})} = 0$, otherwise.

Determining Reasoning-Level Uncertainty via Semantic Relevance. Another source of relevant scenes is the scenes that are semantically similar but occur in different time steps, such as the shooting performed by different people in Fig. 3. Since the setting in $V$ is confined, if the knowledge of $f$ can lead to certain predictions, $f$ shall produce consistent predictions when seeing similar scenes. Based on this intuition, we use a pretrained vision feature extractor $g$ to extract abstract semantic features for each segment. The semantic similarity of any pair of segments $v_i$ and $v_j$ is computed by cosine similarity $\text{sim}(v_i, v_j) = \cos\left(\frac{e_i \cdot e_j}{||e_i|| \cdot ||e_j||}\right)$, where $e_i$ and $e_j$ are the corresponding features for $v_i$ and $v_j$ extracted by $g$. Given any $v_j$, we compute its semantic similarity with all other segments and retrieve the top $K$ most similar scenes, denoted as a set $S_j = [v_j^{(1)}, \cdots, v_j^{(K)}]$. Correspondingly, we have a set of anomaly scores $\tilde{Y}_j$ from $f$ for $S_j$, where $\tilde{Y}_j = \{\tilde{y}_j^{(1)}, \cdots, \tilde{y}_j^{(K)}\}$ is a set of $K$ binary variables. In practice, we set $K$ to be linearly dependent on $h$, e.g., $K = 0.15 \cdot h$. Because of such setting, we use entropy to quantify the reasoning-level uncertainty: firstly, we transform $\tilde{Y}_j$ as a normalized probability distribution $\mathbf{p}_j = [p_j^{(0)}, p_j^{(1)}]$ by

$$p_j^{(0)} = \frac{1}{K} \sum_{k=1}^{K} \mathbb{1}[\tilde{y}_j^{(k)} = 0], \quad p_j^{(1)} = \frac{1}{K} \sum_{k=1}^{K} \mathbb{1}[\tilde{y}_j^{(k)} = 1], \tag{3}$$

where $p_j^{(0)}$ denotes the ratio of normal scenes in $S_j$ judged by $f$ and $p_j^{(1)}$ denotes the ratio of anomalous scenes. After that, we compute the corresponding information entropy, denoted as

$$H(\mathbf{p}_j) = -[p_j^{(0)} \log p_j^{(0)} + p_j^{(1)} \log p_j^{(1)}]. \tag{4}$$

We set a threshold $\delta$ (see details in Sec. A.4.2) to decide whether $f$ is uncertain about $v_j$ denoted by

$$U_j^{(\text{sem})} = \mathbb{1}\{H(\mathbf{p}_j) > \delta\}, \tag{5}$$

where $U_j^{(\text{sem})} = 1$ indicates the existence of uncertainty due to a relatively large entropy value.

### 3.3 UNCERTAINTY MITIGATION BY COLLECTIVE INTELLIGENCE

If for any given $v_j$, we have either $U_j^{(\text{tem})} = 1$ or $U_j^{(\text{sem})} = 1$, UNA will reconsider the decision made by $f$. The uncertainty arises because $f$ is provided with too little evidence, which cannot self-correct reliably (we verify that by experimental results in Table 4 with self-correction mechanisms such as self-reflection (Liu et al., 2024b) and skeleton-of-thoughts (Ning et al., 2024)

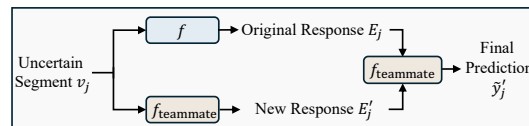

Figure 4: UNA includes a cooperation-based collective intelligence to resolve uncertain predictions.

as baselines). Thus, UNA expands the evidence space to resolve such uncertainty by a collective intelligence strategy ensembling different reasoning paths from different perspectives. W.l.o.g., we illustrate this process with a team with 2 VLMs shown in Fig. 4. In this team, one agent $f_{\text{teammate}}$ works as the teammate of $f$ to review $v_j$ as well and output another language-based explanation $E'_j$ from its own knowledge. After that, $f_{\text{teammate}}$ takes in both opinions, i.e., $E_j$ from $f$ and $E'_j$ from $f_{\text{teammate}}$ along with the visual input $v_j$ to make a final judgment on the given scene. The used prompt template is illustrated in Fig. 5. We denote the new score after mitigation as:

$$\tilde{y}'_j = f_{\text{teammate}}(v_j, E_j, E'_j; \xi), \tag{6}$$

where $\xi$ denotes the used prompt template in Fig. 5, and $\tilde{y}'_j$ is the new prediction after mitigation.

**Discussion.** In Eq. (6), each agent follows its own reasoning trajectory during generation, so their outputs can be regarded as complementary perspectives. Intuitively, this mirrors collective intelligence (Woolley et al., 2010) in human decision-making: independent individuals may reach different conclusions on ambiguous evidence, but their combined judgment often yields a more reliable outcome. From a Bayesian perspective, when we incorporate multiple independent evidence sources from different agents, posterior uncertainty reduces. Please refer to Sec. A.2 in Appendix for a detailed analysis on that if interested.

In addition to the cooperation-based collective intelligence proposed in this paper, other candidates include voting-based and debate-based ones. Compared to that, our mechanism is effective and cost-saving. Due to the space limit, we will provide details of those two variants and the advantages of ours in Sec. A.3 in Appendix.

**VAD by UNA.** Based on the updated initial score, we can obtain a new set of frame-level scores by smoothing techniques developed by previous literature (discussed in

> You are the model.
> **Model Description:** You are designed to do binary classification. The input is a sequence of video frames for identifying whether there is an anomaly in the video. You need to output the class label, i.e., an integer in the set $\{0, 1\}$. 0 represents normal video, and 1 represents abnormal video. Please follow the instruction below to make a conclusion.
> **Instruction:** There are two analyses below, please conclude your answer to 'Is there any anomaly in the video?' in 'Yes, there is an anomaly' or 'No, there is no anomaly' by watching the video carefully and pondering whether these two opinions are consistent.
> **Opinion 1:** [Original Explanation $E_j$ from $f$]
> **Opinion 2:** [New Opinion $E'_j$ from $f_{\text{teammate}}$]
> **Input:** [Video Frames]
> Please give your output strictly in the following format:
> • **New Analysis and Conclusion**: [Please provide a new analysis with these two opinions and make a conclusion.]
> • **Output**: [ONLY the integer class label]
> Please ONLY reply according to this format. Don't give me any other words.

Figure 5: The teammate in UNA uses this prompt template to aggregate two reasoning paths to resolve uncertain scenarios.

Sec. 3.1) and explanations for the test video $V$. Please refer to Alg. 1 in Appendix for the whole procedure if interested. The new explanation will incorporate an uncertainty-aware viewpoint to explain the occurrence of an anomaly, which is illustrated with a case study in Fig. 9.

## 4 EXPERIMENTS AND RESULTS

Given the methodology introduced above, we would like to further present an evaluation of UNA with a focus on the following questions: (Q1) Does the proposed uncertainty-aware framework enhance the effectiveness of VLMs for VAD? (Q2) Is the design of uncertainty identification and mitigation in UNA reasonable? (Q3) How does the proposed design improve the explainability of VLMs for VAD?

### 4.1 EXPERIMENTAL SETTINGS

**Datasets**. Following pioneering VLM-based VAD studies (Zanella et al., 2024; Ye et al., 2025), We evaluate UNA on two large-scale VAD datasets, UCF-Crime (Sultani et al., 2018) and XD-Violence (Wu et al., 2020), as other VLM-based VAD studies (Zanella et al., 2024; Ye et al., 2025) do. Details of the datasets are in Sec. A.4.1 in Appendix.

**Metrics**. We mainly adopt the Area Under the Curve (AUC) frame-level Receiver Operating Characteristic (ROC) curve for VAD performance evaluation, following representative VLM for VAD works (Zanella et al., 2024; Zhang et al., 2024a; Ye et al., 2025).

**Implementation of UNA**. We use open-source VLMs from the InternVL (Chen et al., 2024b) and Qwen-VL (Wang et al., 2024) families to build UNA. We do not conduct any fine-tuning on the used models and adopt prompts trained from verbalized learning as (Ye et al., 2025) does. Due to the space limit, please refer to Sec. A.4.2 in Appendix for the setting of other hyperparameters, including window size $w$, the number of similar scenes $K$, vision feature extractor $g$, and the entropy threshold $\delta$.

**Baselines**. In this paper, we follow the latest literature to classify used baselines into two categories: non-explainable approaches including BODS (Wang & Cherian, 2019), Chen et al. (Chen et al., 2024a), CLAWS (Zaheer et al., 2020), CLIP-TSA (Joo et al., 2023), DYAN-NET (Thakare et al., 2023b), GCL (Zaheer et al., 2022), GCN (Zhong et al., 2019), GODS (Wang & Cherian, 2019), Hasan et al. (Hasan et al., 2016), Lu et al. (Lu et al., 2013), MGFN (Chen et al., 2023), MIST (Feng et al., 2021), MSL (Li et al., 2022b), OVVAD (Wu et al., 2024a), RareAnom (Thakare et al., 2023a), RTFM (Tian et al., 2021), S3R (Wu et al., 2022), SSRL (Li et al., 2022a), Sultani et al. (Sultani et al., 2018), TPWNG (Yang et al., 2024b), Tur el al. (Tur et al., 2023), VadCLIP (Wu et al., 2024b), and Wu et al. (Wu et al., 2020), and explainable ones including Holmes-VAD (Zhang et al., 2024a), LAVAD (Zanella et al., 2024), LLAVA-1.5 (Liu et al., 2024a), VADor (Lv & Sun, 2024), VERA (Ye et al., 2025), Zero-Shot CLIP (Zanella et al., 2024), and Zero-Shot ImageBind (Girdhar et al., 2023). Non-explainable ones can only output numerical prediction, while explainable ones can generate predictions and explanations.

**Comparison Setting**. In this paper, we focus on the most challenging VAD setting for VLMs, where no instruction tuning datasets are available for VLMs for fine-tuning. Consequently, all VLM-based baselines are evaluated without fine-tuning.

Table 1: Comparison of VAD performance on UCF-Crime.

| Method | AUC (%) |
|---|---|
| *Non-explainable VAD Methods* | |
| Chen et al. | 86.83 |
| CLAWS | 83.03 |
| CLIP-TSA | 87.58 |
| DYANNET | 84.50 |
| GCL | 79.84 |
| GCN | 82.12 |
| GODS | 70.46 |
| MGFN | 86.98 |
| MIST | 82.30 |
| MSL | 85.62 |
| OVVAD | 86.40 |
| RTFM | 84.30 |
| S3R | 85.99 |
| SSRL | 87.43 |
| Sultani et al. | 77.92 |
| TPWNG | 87.79 |
| Tur el al. | 66.85 |
| VadCLIP | 88.02 |
| Wu et al. | 82.44 |
| *Explainable VAD Methods* | |
| Holmes-VAD | 84.61 |
| LAVAD | 80.28 |
| LLAVA-1.5 | 72.84 |
| VADor | 85.90 |
| VERA | 86.55 |
| ZS CLIP | 53.16 |
| ZS IMAGEBIND-I | 53.65 |
| ZS IMAGEBIND-V | 55.78 |
| UNA | **88.27** |

## 4.2 COMPARISON WITH STATE-OF-THE-ART METHODS

We present the results of comparison with state-of-the-art VAD approaches first to address (Q1). First, on the challenging UCF-Crime benchmark, with an extra consideration on the uncertain scenarios in VAD reasoning, UNA achieves the highest AUC among all VAD methods on UCF-Crime, as Table 1 shows. This is impressive because: (1) UNA makes frozen VLMs surpass the performance of the best task-specific non-explainable VAD method for the first time (88.27% vs 88.02%), which proves the great potential of VLMs applied in VAD. (2) UNA obtains better performance compared to other VLM-based methods by having an extra teammate to tackle the reasoning-level uncertainty in the base VLM model, which is the main difference between UNA and others. This validates the necessity of an uncertainty-aware perspective in VLMs for VAD.

This excellent performance is also validated in Table 2 for another benchmark XD-Violence (note that we report average precision

Table 2: Comparison of VAD performance on XD-Violence.

| Method | AUC (%) |
|---|---|
| *Non-Explainable VAD Methods* | |
| BODS | 57.32 |
| GODS | 61.56 |
| Hasan et al. | 50.32 |
| Lu et al. | 53.56 |
| RareAnom | 68.33 |
| *Explainable VAD Methods* | |
| LAVAD | 85.36 |
| LLAVA-1.5 | 79.62 |
| ZS CLIP | 38.21 |
| ZS IMAGEBIND-I | 58.81 |
| ZS IMAGEBIND-V | 55.06 |
| VERA | 88.26 |
| UNA | **91.11** |

results for XD-Violence in Sec. A.5.1 in Appendix). To illustrate, UNA further improves the performance of explainable VAD methods to a 91.11% AUC in XD-Violence, which improves 2.85% compared to the second best. To conclude, an uncertainty-aware framework will highly enhance the effectiveness of VLMs for VAD.

Table 3: Influence of uncertainty identification.

| Operation | AUC (%) |
|---|---|
| InternVL2-8B (Base Model $f$) | 86.33 |
| + Option 1 (Token Probability) | 86.58 (+0.25) |
| + Option 2 (Self-Consistency) | 87.04 (+0.71) |
| + Uncertainty via Temporal Relevance | 87.15 (+0.82) |
| + Uncertainty via Semantic Relevance | 87.31 (+0.98) |
| + Uncertainty via Eq. (2) and Eq. (5) | 87.57 (+1.24) |

Table 4: Influence of uncertainty mitigation.

| Operation | AUC (%) |
|---|---|
| InternVL2-8B | 86.33 |
| + Self-Reflection | 85.03 (-1.30) |
| + Skeleton-of-Thoughts | 85.00 (-1.33) |
| + Voting | 86.89 (+0.56) |
| + Debate | 87.00 (+0.67) |
| + Cooperation by Eq. (6) | 87.57 (+1.24) |

### 4.3 ABLATION STUDIES FOR METHODOLOGY DESIGN

We now investigate the reasonableness of the design of UNA to address (Q2) by ablation studies. W.l.o.g., we conduct experiments on the UCF-Crime dataset, using InternVL2-8B as $f$ in UNA.

**Uncertainty Identification Design**. We first validate the design of uncertainty identification. As Table 3 (mitigation is the same for all variants) shows, if we replace the uncertainty test with Option 1, the increase of AUC in UCF-Crime is only 0.25%, which indicates the ineffectiveness of token probability in determining uncertain scenarios. Option 2 can improve the result, but the extra compute ($N = 3$) is costly. Meanwhile, we find that individual use of the proposed temporal and semantic relevance can improve the AUC more with absolute values of 0.82% and 0.98%, respectively, which is better than the baselines. When combined, they yield a 1.24% performance improvement, indicating that they are complementary and mutually beneficial for quantifying the reasoning-level uncertainty.

**Uncertainty Mitigation Design**. We now verify the proposed design for uncertainty mitigation, as shown in Table 4. The baselines are self-improving mechanisms, including self-reflection (Liu et al., 2024b) and skeleton-of-thoughts (Ning et al., 2024), which prompts the base model $f$ to think again by itself when uncertainty is identified. However, they only make VAD performance worse for only using one perspective. On the contrary, when we introduce collective intelligence by any mechanism discussed in Sec. 3.3, the VAD performance will increase. These results validate the necessity of introducing collective intelligence in resolving reasoning-level uncertainty in an individual VLM. Furthermore, because of the ensemble of different reasoning paths, the proposed cooperation-based collective intelligence improves the performance most, which verifies the reasonableness of UNA.

**Selection of Teammate in Cooperation**. After that, we investigate the influence of the teammate architecture in UNA. As Fig. 6 shows, first, using $f$ itself as the teammate improves slightly because uncertainty cannot be self-corrected. Meanwhile, having a different VLM as the teammate yields better improvement. These results indicate the benefit brought by collective intelligence – a different perspective helps decrease reasoning-level uncertainty in $f$. Secondly, the mitigation ability of the teammate is related to its reasoning ability in VAD. Qwen2-VL-7B, InternVL-3, and InternVL-2.5 are weak

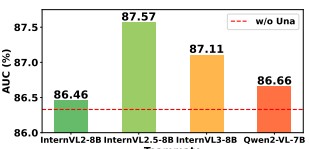

Figure 6: Influence of teammate architectures in UNA.

to strong reasoning models for VAD (shown in Fig. 7), and when we adopt them as $f_{\text{teammate}}$, the performance improves the most with InternVL-2.5 and the less with Qwen2-VL-7B. Thus, the stronger reasoning ability of the teammate in VAD, the better it can improve $f$.

**Generalizability across Different Architectures**. We further test the generalizability on the base model $f$ with different architectures. From Fig. 7, we find that in general using a single VLM usually has unsatisfactory performance due to its poor reasoning, which is from its limited knowledge in VAD. However, their limitation can be tackled by the employment of UNA. For example, when $f$ is InternVL3-8B, the collective intelligence in UNA allows it to resort to other VLM agent to handle uncertainty, making the AUC improve to 88.27% (the SOTA performance reported in Table 1) from 83.72%. In addition, the proposed

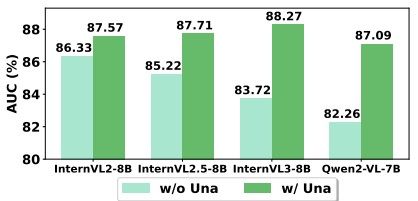

Figure 7: UNA is generalizable across different base models.

UNA framework is generalizable across different models: when we change the base model from the InternVL family to Qwen2-VL-7B, UNA improves the AUC to 87.09% from 82.26%. Such an improvement in AUC is quite impressive (+4.83%), and it is comparable to that of InternVL3-8B. These results in Fig. 7 prove that an uncertainty-aware mechanism is essential for VLM in VAD.

**Generalizability across Different Prompts**. UNA is insensitive to prompt variations. W.l.o.g., we test UNA with two different prompts obtained from VERA (Ye et al., 2025), using InternVL2-8B as $f$. They make $f$ have an 83.29% and 83.49% in AUC, respectively, indicating a gap of 3% with the best one. After applying UNA on $f$ with these variants, the performance on UCF-Crime increases to 86.45% and 86.06%, respectively, which narrows the gap with the best one into 1.5% in AUC. These results show that UNA can help achieve robust improvement for an individual VLM with different prompts.

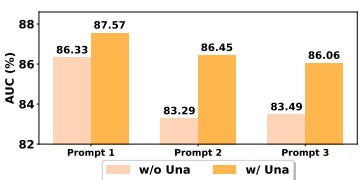

Figure 8: UNA is generalizable across different prompts used by the base model.

**Discussion: Scaling and Computational Cost**. We further use InternVL3-8B as $f$ and add InternVL2-8B, InternVL3-8B, and Qwen2-VL-7B gradually to build a team from 2 VLMs to 4 VLMs. We average the re-evaluation score with 3 or more agents for the final prediction. The results in Table 5 show that

Table 5: UNA can improve by scaling up teammates.

| # VLMs | 2 | 3 | 4 |
|---|---|---|---|
| AUC (%) | 88.27 | 88.81 | 88.72 |

increasing the number of models consistently improves AUC, demonstrating that scaling enhances performance. While the gains converge as more agents are added, UNA achieves a stably high level of AUC, confirming that collaborative scaling is effective.

We understand that computational cost may be a concern in specific scenarios. In the default setting with 2 VLMs, in the worst case, we need twice the computation time given the same resource. In our hardware environment, the FPS is 2.52 (GFLOPs are $3.2 \times 10^3$) for a single uncertain segment, which is twice (a half) that of a single agent, with running time reported in Sec A.4.4. A future direction to improve efficiency is token reduction (Zhang et al.) to send multiple requests in parallel.

## 4.4 QUALITATIVE RESULTS AND CASE STUDIES

W.l.o.g., we now answer (Q3) with a showcase of how UNA helps improve the base model in uncertain cases with an instance shown in Fig. 9 from a long, difficult burglary video (Burglary076_x264) in UCF-Crime with a mix of normal and abnormal events (Please refer to Sec. A.5.2 for more details). The video

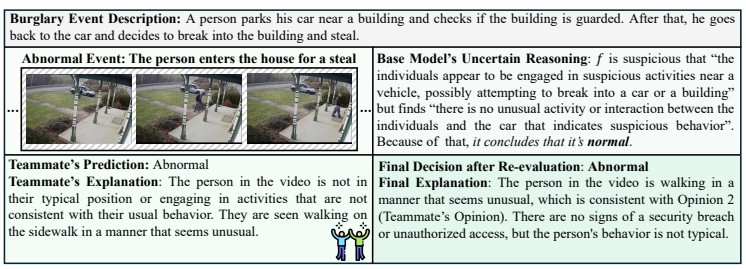

Figure 9: UNA utilizes an alternative viewpoint from the teammate to mitigate the reasoning-level uncertainty.

looks normal at first when a person parks the car on a lawn and waits at the front door of the building, but he later decides to break into the building to steal. The base model $f$ exhibits uncertainty throughout the prediction process. Due to the ambiguity, the base model $f$ (InternVL2-8B) wrongly pays attention to the interaction between the person and the car and thinks it is normal. UNA identifies uncertainty in this reasoning via Eq. (5). Meanwhile, in UNA the teammate (InternVL2.5-8B) provides another opinion by focusing on the action of walking into the building of that person. Finally, UNA corrects the uncertain prediction by the cooperation mechanism in Eq. (6) and outputs an accurate, affirmative explanation that it is an anomaly for his abnormal walking. This validates that collective intelligence is necessary in improving the explainability of VLM in VAD.

## 5 CONCLUDING REMARKS

This paper proposes a new uncertainty-aware perspective for employing VLMs for VAD. Specifically, we introduce an UNA framework with uncertainty identification and mitigation to reduce the potential errors caused by reasoning-level uncertainty: (1) UNA identifies reasoning-level uncertainty in VLM for VAD via inspecting its prediction consistency among relevant scenes from temporal and semantic information. (2) UNA mitigates the uncertain cases in VAD by relying on collective intelligence obtained from VLM cooperation. Experimental results validate the necessity of modeling reasoning-level uncertainty in VAD and the effectiveness of introducing collective intelligence.

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

# A APPENDIX

## A.1 PSEUDOCODES OF UNA

Alg. 1 shows the pseudocodes of UNA. Firstly, as previous methods do, UNA generates the segment-level scores by inputting each segment into the deployed base model $f$. After that, UNA conducts the reasoning-level uncertainty tests on each segment $v_j$ with the temporal relevance and the semantic relevance. If an uncertain segment is found, UNA will introduce a teammate to address the uncertainty. Lastly, the rectified initial scores are put to a smoothing technique to output the desired frame-level anomaly scores for all frames.

---

**Algorithm 1:** Uncertainty-aware VLMs for VAD by UNA

---

**Input**: A set of video segments $\{v_j\}_{j=1}^h$, a VLM model $f$ employed in VAD, and the teammate $f_{\text{teammate}}$.

# Obtaining Segment-Level Initial Scores

**for** $j \leq h$ **do**

  Input $v_j$ into $f$ to compute an initial anomaly score $\tilde{y}_j$

**end for**

# Uncertainty-Aware VAD

**for** $j \leq h$ **do**

  **if** $f$ is uncertain on $v_j$ decided by the reasoning-level uncertainty test by Eq. (2) or Eq. (5) **then**

    Introduce a teammate $f_{\text{teammate}}$ to resolve the uncertainty by Eq. (6)

  **end if**

# Obtaining Frame-Level Scores

Apply the smoothing technique by (Ye et al., 2025) to refine the new coarse-grained scores into the fine-grained scores $\hat{Y}$

**Return** $\hat{Y}$

---

## A.2 THEORETICAL ANALYSIS ON THE EFFECTIVENESS OF UNA

Let $Y$ denote the true binary label of a segment and let $E_1$ be the evidence (explanation/perspective) provided by the base model $f$. By Bayes' rule the posterior given $E_1$ is

$$P(Y \mid E_1) = \frac{P(E_1 \mid Y)\,P(Y)}{P(E_1)}. \tag{7}$$

If we incorporate an additional independent perspective $E_2$ from the teammate $f_{\text{teammate}}$. Since they generate the evidence (explanation) independently, we can assume conditional independence of the evidence given $Y$, the joint posterior is

$$P(Y \mid E_1, E_2) \propto P(E_1 \mid Y)\,P(E_2 \mid Y)\,P(Y). \tag{8}$$

Because adding an independent likelihood term concentrates the posterior, the posterior uncertainty cannot increase. In terms of variance, we have

$$\mathrm{Var}\big[Y \mid E_1, E_2\big] \leq \mathrm{Var}\big[Y \mid E_1\big], \tag{9}$$

with equality iff $E_2$ is conditionally independent of $Y$ given $E_1$ (i.e., $E_2$ carries no additional information about $Y$ beyond $E_1$). Equivalently, in information-theoretic terms,

$$H\big(Y \mid E_1, E_2\big) \leq H\big(Y \mid E_1\big), \tag{10}$$

since conditioning on more informative, independent data reduces the conditional entropy $H(\cdot)$. Thus, adding an independent teammate reduces posterior uncertainty.

## A.3 DISCUSSION ON MITIGATION DESIGN

We illustrate the collective intelligence mechanisms implemented by voting, debate, cooperation in Fig. 10. The details of voting-based one and debate-based one are as follows:

Variant 1: Voting-Based Collective Intelligence. In this variant, we have two additional VLMs $f_{\text{voter2}}$ and $f_{\text{voter3}}$ applied in the uncertain case. When UNA detects uncertainty in $f$ for a given

Figure 10: We can implement collective intelligence mechanisms with voting, debate, and cooperation. The proposed cooperation-based one is more suitable for VLM-based VAD because it allows different perspectives to be fused and requires less compute.

segment $v_j$, it will feed $v_j$ into each VLM and it makes a decision independently on whether $v_j$ contains anomalies. The majority class predicted by them is used as the final prediction for the uncertain case. The prompt used by each voter is detailed in Sec. A.4.5.

Variant 2: Debate-Based Collective Intelligence. In this variant, one VLM-based agent serves as a debater $f_{\text{debate}}$ against $f$'s opinion when disagreement happens for detected uncertain scenarios, and another agent serves as a moderator $f_{\text{moderate}}$ to judge which agent is correct. In this variant, when two opinions are presented to the moderator, the moderator makes the final decision. This is similar to the method by (Liang et al., 2024). The prompt used by the moderator is detailed in Sec. A.4.5.

**Discussion**. The proposed cooperation-based one detailed in Sec. 3.3 outperforms these variants for VLMs in VAD as follows. First, if we apply a voting-based one on VAD, it simply asks agents to conduct VAD independently and aggregate the results with a majority vote. However, such collaboration lacks information flow between different types of world knowledge as UNA does. Thus, the reasoning-level uncertainty cannot be effectively relieved by voting. Second, debate-based one involves a moderator which requires additional computation. It requires an extra agent to fuse different responses, which costs additional compute. Compared to that, the cooperation-based collective intelligence amends reasoning-level uncertainty by utilizing only one teammate to perceive two views and reduce errors made by the single agent $f$ in uncertain cases. This analysis is verified by the ablation study results we previously showed in Table 3.

## A.4    EXPERIMENT AND IMPLEMENTATION DETAILS

### A.4.1    DATASET DETAILS

We conduct experiments on two large-scale VAD datasets: (1) UCF-Crime (Sultani et al., 2018) and (2) XD-Violence (Wu et al., 2020). The details are as follows:

- **UCF-Crime** dataset is collected from real-world surveillance videos (128-hour long in total), covering crime-related anomalies including abuse, arrest, arson, assault, burglary, explosion, fighting, road accident, robbery, shoplifting, shooting, stealing, and vandalism. The training set has 1610 videos (810 abnormal ones and 800 normal ones), while the test set has 290 videos (140 abnormal ones and 150 normal ones). The total number of test frames is over 1 million (1,111,808), and abnormal frames account for 7.92%. The average duration of a test video is 2.13 minutes, which is relatively long compared to common video datasets and serves as an important benchmark.

- **XD-Violence** is another representative large-scale (217-hour long in total) VAD dataset with 6 anomaly categories, i.e., abuse, car accident, explosion, fighting, riot, and shooting, which defines anomalous events as the ones related to violence. This dataset is collected from movies and YouTube videos. It has 3954 training videos and 800 test videos (500 abnormal ones and 300 normal ones). The total number of test frames is over 2 million (2,335,801), and abnormal frames account for 23.07%. The average duration of a test video is 1.62 minutes.

### A.4.2    HYPERPARAMETER SETTING

During inference, following (Zanella et al., 2024; Ye et al., 2025), the interval between each segment center $d$ is 16 frames, the number of sampled frames is $\kappa = 8$ for each segment, and each segment focuses on a 10-second glimpse around the center. When obtaining similar scenes with temporal

dynamics, the window size $w = 5$. When computing the semantic similarity, we use ImageBind as the feature extractor $g$ and the number of semantically similar scenes $K$ depends on the total number of segments $h$ in each test video $V$. We set $K$ to $(0.15 \cdot h)$. The threshold $\delta$ is set according to the distribution of entropy values across all segments, i.e., $[H(\mathbf{p}_1), \cdots, H(\mathbf{p}_h)]$, and can be set to the $\varsigma$-th percentile of this distribution. In this way, we will consider top $\varsigma\%$ video segments in the entropy list as the segments $f$ is uncertain about. By default, we set $\delta$ as the median of the entropy value list.

### A.4.3 SENSITIVITY TEST FOR HYPERPARAMETERS

For key hyperparameters including $w$, $K$, and $\delta$, w.l.o.g., we conduct sensitivity tests on them on the UCF-Crime dataset, with InternVL3-8B as $f$ and InternVL2-8B as $f_{\text{teammate}}$. The results are as follows, with other hyperparameters fixed when alternating one.

**Sensitivity Test for** $w$. From Table 6, the performance of UNA is stable with a varying $w$ from 3 to 11. This result shows that the VAD performance is insensitive to $w$. Thus, we can set $w = 5$ in general cases.

Table 6: Influence of the window size $w$ on AUC (%).

| $w$ | 3 | 5 | 7 | 9 | 11 |
|-----|-------|-------|-------|-------|-------|
| AUC | 88.37 | 88.27 | 88.28 | 88.24 | 88.16 |

**Sensitivity Test for** $K$. Table 7 shows that if we do not use semantically similar scenes (i.e., setting $K = 0$), the performance will drop to 84.57%, which demonstrates the necessity of introducing scenes with similar semantic meaning for resolving uncertain scenarios. Meanwhile, when $K$ is alternated in a reasonable range around $0.05 \cdot h$ to $0.15 \cdot h$, the performance will be stable and kept high. If we set $K$ to a relatively huge number $0.20 \cdot h$, the performance will drop slightly. Thus, selecting $0.15 \cdot h$ for $K$ is generally a good choice.

Table 7: Influence of the number of appearance-similar scenes $K$ on AUC (%).

| $K$ | 0 | $0.05 \cdot h$ | $0.10 \cdot h$ | $0.15 \cdot h$ | $0.20 \cdot h$ |
|-----|-------|-------|-------|-------|-------|
| AUC | 84.57 | 88.14 | 88.30 | 87.97 | 86.54 |

**Sensitivity Test for** $\delta$. The choice of $\delta$ depends on all entropy values obtained from each segment. We alternate $\delta$ as the 10th percentile, the 30th percentile, the 50th percentile (Median), the 70th percentile, and the 90th percentile of the given entropy values. The results in Table 8 show that when $\delta$ is a relatively small number (less than the median), the performance will have a relatively high AUC. However, when it becomes relatively large which makes scenes with similar semantics hardly included, the AUC will drop dramatically to 84.58%. Thus, including semantically similar scenes is necessary, and setting $\delta$ to the median of the entropy list is a generally good practice.

Table 8: Influence of the threshold $\delta$ on AUC (%).

| $\delta$ | 10th percentile | 30th percentile | Median | 70th percentile | 90th percentile |
|-----|-------|-------|-------|-------|-------|
| AUC | 87.27 | 87.89 | 88.27 | 85.80 | 84.58 |

### A.4.4 RUNNING TIME

This paper requires deploying vision-language models for generating responses given certain prompts (detailed in Sec. A.4.5). Specifically, for a given segment, the proposed framework requests $L$ queries in total to the multi-agent collaboration system with $L$ agents. In our hardware environment with NVIDIA A6000 GPUs, a query costs around 4.29 seconds for a model from the InternVL family to respond. As a result, for a single VLM-based system, in a representative dataset like UCF-Crime which costs an average of 239 queries per video, the average inference time per video is 1027s.

### A.4.5 PROMPTS USED IN INFERENCE

**Prompt for Voting-Based Collective Intelligence**. In the voting-based collaboration, each agent performs the same operation for predicting the existence of an anomaly in the same segment. Thus,

they will use the same prompt template as follows. We follow (Ye et al., 2025) to prompt all voters for video anomaly detection for this prompt template has been found to be successfully useful for VLMs in VAD:

```
  You are the model.

                    ** Model Description:  **

You are designed to do binary classification.  The input is a
sequence of video frames for identifying whether there is an
anomaly in the video; you need to output the class label, i.e.,
an integer in the set 0, 1.  0 represents normal video, and 1
represents abnormal video.  Please answer the prompt questions.

                    ** Prompt Questions:  **

Answer the following questions based on what you see from the
video frames and provide an explanation in one sentence.

                        [Guiding Question]

Based on the analysis above, please conclude your answer to 'Is
there any anomaly in the video?'  in 'Yes, there is an anomaly' or
'No, there is no anomaly'.

                        ** Input:  **
                        [Visual Data]

Please give your output strictly in the following format:
Answers to Prompt Questions:
[Provide your analysis by answering the questions listed in Prompt
Questions.]
Output:
[ONLY the integer class label; make necessary assumptions if
needed]
Please ONLY reply according to this format, don't give me any
other words.
```

With this template used in the implementation, the corresponding input video segment will replace the [Visual Data] block, and the guiding question block [Guiding Questions] will be replaced by the ones learned from the verbalized learning mechanism in VERA (Ye et al., 2025). For example, the one learned by the base model $f$ with InternVL2 is "1. Are there any people in the video who are not in their typical positions or engaging in activities that are not consistent with their usual behavior? 2. Are there any vehicles in the video that are not in their typical positions or being used in a way that is not consistent with their usual function? 3. Are there any objects in the video that are not in their typical positions or being used in a way that is not consistent with their usual function? 4. Is there any visible damage or unusual movement in the video that indicates an anomaly? 5. Are there any unusual sounds or noises in the video that suggest an anomaly?"

**Prompt for Debate-Based Collective Intelligence**. In this mechanism, the base model and the opponent use the same prompt shown in the voting-based collaboration for making their own predictions. We need to introduce a new prompt for the moderator, which is as follows:

```
  You are the moderator.

                    ** Moderator Descriptions:  **

You are designed to do binary classification on whether an anomaly
exists in the given video input.  The input is a sequence of video
frames for identifying whether there is an anomaly in the video.
You need to output the class label, i.e., an integer in the set
```

```
0, 1.  0 represents normal video, and 1 represents abnormal video.
Please follow the instruction below to make a conclusion.

                        ** Instruction:  **

There are two debaters involved in debating whether an anomaly
exists.  They will present their analyses and discuss their
perspectives.  Please evaluate both sides' analyses and decide
which one is correct.  Please conclude your answer to 'Is there
any anomaly in the video?'  by what you see in the video.

                    ** Debater 1's Argument:  **
                        [Debater's Analysis]
                    ** Debater 2's Argument:  **
                        [Debater's Analysis]
                           ** Input:  **
                          [Visual Data]

Please give your output strictly in the following format:
New Analysis and Conclusion:
[Please provide a new analysis with these two opinions and make a
conclusion.]
Output:
[ONLY the integer class label]
Please ONLY reply according to this format, don't give me any
other words.
```

During the implementation of this mechanism, the argument held by each agent in the debate will be put into the blocks of [Debater's Analysis].

## A.5 ADDITIONAL RESULTS

### A.5.1 COMPARISON TO THE STATE-OF-THE-ART METHODS ON XD-VIOLENCE MEASURED BY AP

We further show the comparison results measured by average precision (AP), i.e., the area under the frame-level precision-recall curve, on XD-Violence in Table 9. In case we may wonder, the gap on AP by UNA and other methods like Holmes-VAD and CLIP-TSA is understandable because they use full training data to train the model, as pointed out by (Ye et al., 2025; Wu et al., 2024a). In a fair comparison setting where models are not trained with full data, from Table 9 we find that UNA has a 6.59% increase on AP compared to VERA, which again demonstrates that UNA can improve VLM-based methods with an uncertainty-aware mechanism.

### A.5.2 CASE STUDY DETAILS

We would like to provide more details on the case study we have in Sec. 4.4. We showcase another mitigation case with a normal segment in the video, as Fig. 11 illustrates. In the beginning, the video looks normal at first when a person parks the car on a lawn and waits at the front door of the building. Due to the limitation of the prompt, despite finding no evidence for an anomaly, the base model $f$ (InternVL2-8B) wrongly concludes that the scene is abnormal. Such uncertainty is found by UNA, and it introduces the perspective of a teammate (InternVL2.5-8B) to re-evaluate and output a final prediction. Since both opinions agree that no unusual activities occur, the final prediction is rectified as normal.

Table 9: Comparison of performance on XD-Violence measured by AP (%). † indicates VAD methods are trained on entire training frames.

| Method | AUC |
|---|---|
| *Non-Explainable VAD Methods* | |
| Wu et al.[†] | 78.64 |
| OVVAD | 66.53 |
| RTFM[†] | 77.81 |
| MSL[†] | 78.58 |
| MGFN[†] | 80.11 |
| CLIP-TSA[†] | **82.19** |
| *Explainable VAD Methods* | |
| Holmes-VAD[†] | **84.96** |
| LAVAD | 62.01 |
| ZS CLIP | 17.83 |
| ZS IMAGEBIND-I | 27.25 |
| ZS IMAGEBIND-V | 25.36 |
| LLAVA-1.5 | 50.26 |
| VERA | 70.54 |
| UNA | 77.13 |

| | |
|---|---|
| **Burglary Event Description:** A person parks his car near a building and checks if the building is guarded. After that, he goes back to the car and decides to break into the building and steal. | |
| **Normal Event: : A car is parked near a building** | **Base Model's Uncertain Reasoning**: The base model finds that "no unusual activities or unexpected events are occurring in the video". Despite that, in the end, *the base model predicts that this is an anomalous scene*, which counters its reasoning and is inconsistent with its prediction around the temporal window. |
| **Teammate's Prediction:** Normal
**Teammate's Explanation**: : It's normal because "there are no people in the video, and the vehicles and objects are in their typical positions and being used in their usual ways." | **Final Decision after Re-evaluation**: **Normal**
**Final Explanation**: Both opinions agree that there are no unusual activities or unexpected events occurring in the video. The car is parked in a typical location and at a typical time. There is no person walking or interacting with the car. The environment and surroundings appear normal. There are no signs of a security breach or unauthorized access in the video. |

Figure 11: Another exemplary vidoe segment demonstrating how UNA utilizes an alternative viewpoint from the teammate to mitigate the reasoning-level uncertainty.

We also want to show the effectiveness of UNA in obtaining relevant scenes that can help identify whether the base model is uncertain about its prediction with a certain scene. As shown in Fig. 12, w.l.o.g., we take the second scene in Fig. 9 as an example for demonstration. As seen from the left-hand side, the relevant scenes obtained from temporal relevance are closely related to the main scene, which captures the same event that a person is walking from the car and trying to break into a building with a slight change of temporal window. As for the right-hand side, it shows the scenes relevant to the main scene measured by semantic similarity. These scenes do not generally occur close to the main scene in the time domain and share appearance similarities such as a man showing up near the door. By closely observing these scenes, we can find that they are closely related to the main scene, and they will help us judge if the base model is confident because the base model shall maintain consistent predictions among them as they depict similar activities or events. Thus, the design in UNA for obtaining relevant scenes by considering temporal dynamics and semantic similarity will allow us to find useful relevant scenes determining whether the base model has any uncertainty.

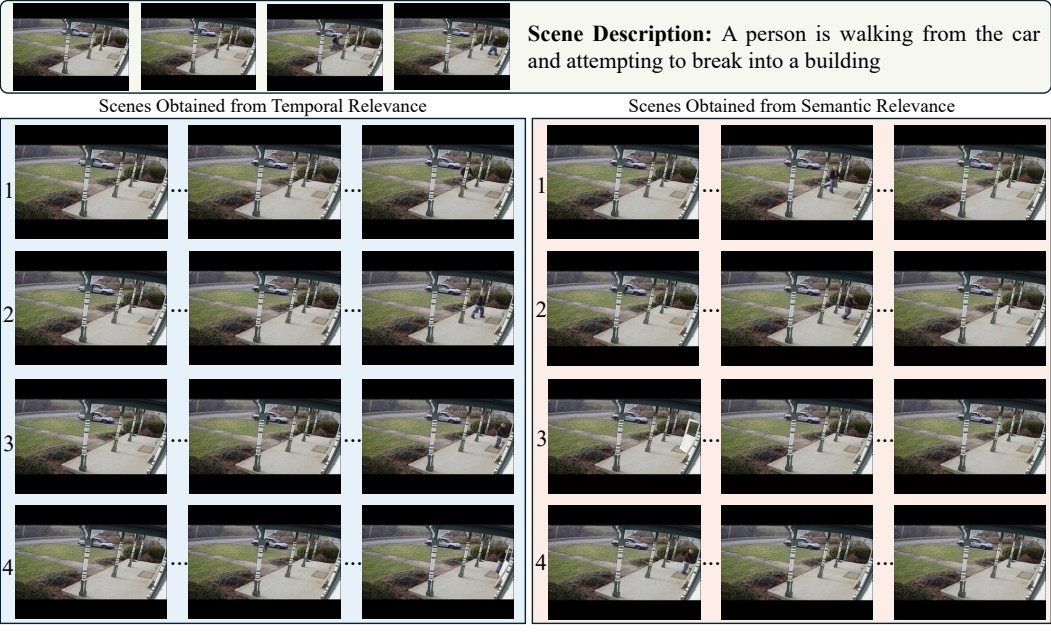

Figure 12: We take "Burglary076_x264" from UCF-Crime as an example to showcase the relevant scenes obtained from UNA via temporal and semantic relevance.

## A.6 THE USE OF LARGE LANGUAGE MODELS (LLMs)

Large language models were used solely for minor polishing during manuscript preparation. They were not used for research ideation, retrieval and discovery, methodology design, experiment implementation, or result analysis.

