# OpenReview forum: "Two are Better than One: Uncertainty-Aware Vision-Language Models for Video Anomaly Detection"
_ICLR.cc/2026/Conference — ICLR 2026 Conference Withdrawn Submission_

### Official Review · Reviewer_88Bo · 2025-10-28

**Soundness:** 2
**Presentation:** 2
**Contribution:** 1
**Rating:** 2
**Confidence:** 4

**Summary:**

This paper introduces to use uncertainty modeling in vision language models to improve video anomaly detection.

The proposed model aims to address the issues of video segment-based methods that ignore the temporal context information for robust anomaly detection.

Experimental results are performed on two benchmark datasets and show improved performance.

**Strengths:**

+ Clearly outline the key aspects of video anomaly detection regarding accuracy, explainability, and generalizability.

+ The paper is somehow well organised, and many figures, plots and tables are presented, visually attractive. Some concepts being clearly explained using figures such as fig 3.

+ Improved performance on both XD-Violence and UCF-Crime datasets.

**Weaknesses:**

Major

- This work is not very well motivated, and the claim “existing methods directly model VAD as natural language generation task but ignore the uncertainty in their reasoning” is questionable, as VLMs and/or LLMs have also been used for video anomaly understanding and reasoning with uncertainty modeling capacity. Why eg transformer-based models, and long-term temporal modeling methods are unable to address the challenges?

- The concept of using uncertainty is unclear (not strongly, clearly and well motivated). What and where are the potential uncertainties? In intro, these concepts are not being clearly explained, and only fig 2 provides some insights regarding uncertainty. How current VAD datasets reflect on the concept of uncertainty. What about the temporal length of each video segment, does this reflect on reasoning-level uncertainty? These experiments and evaluations are not being performed.

- The evaluations are conducted only on two old datasets, it would be better to use some new datasets such as MASD (NeurIPS’24). It would be better to show how the proposed model on both scenario- and anomaly-based evaluations on this dataset. The current evaluations do not reflect on how uncertainty modeling benefits different scenarios and different anomaly types etc. Although the generalizability across architectures is provided, it would be better to show how the generalizability across different scenarios and different anomaly types.

- The newly introduced VAD measurement is impractical due to its hard voting nature (either 0 or 1), it would be better if the authors could explain this concept clearly.

- Equation errors such as Line 251, the calculation inside the right side of equation is the cosine similarity (L2 followed by dot product), but the calculated results is again being passed to cosine function. The reviewer is confused here what is the similarity being used here, as that seems like cos(cos($e_i$, $e_j$)) (cosine being used twice)? Eq 6, the teammate function/model is unclear to reviewer what is the exact setup.

- Lack of justifications or empirical studies on the choice of parameters. Line 6, the threshold value is not being defined, and section A4.2 does not provide any details regarding the justification of using this threshold or the parameter setups and evaluations. How parameter evaluations are being performed are unclear and there is no hyperparameters evaluation.

-  What kind of explainability does this work focus on? The metrics being used in evaluations do not reflect on the explainability of VLMs for VAD, quantitative and qualitative evaluations are needed apart from AUC/ROC to show improved “explainability”.

Minor:

- Some sentences read not very smooth, such as Line 73, and Line 282 due to the heavily use of “with”.

- Some words such as “W.l.o.g.” could be improved.

- In line 256, why 0.15 is being used? This is unclear and no any justification is provided.

**Questions:**

Refer to weaknesses, and also the following questions.

- How the segment-level score is being refined towards the frame-level anomaly score?

- In line 135, are the video frames being sampled (some frames being dropped)? Would it potentially drop anomaly frames if the step size is bigger? Also in Line 140, it says “uniformly sampling”.

- Why use hard voting (either 0 or 1) rather than soft voting (eg with probability scores within 0 and 1, such as 0.5, etc)?

- What is the relationship between h segments (line 139-140) and F (line 135)?

- Line 157, “constrained to isolated segments”, are there overlapping frames between consecutive video segments (any overlapping frames in each pair of consecutive segments, Line 224-226)?

---

### Official Review · Reviewer_XX8w · 2025-10-29

**Soundness:** 3
**Presentation:** 2
**Contribution:** 2
**Rating:** 4
**Confidence:** 3

**Summary:**

The paper introduces UNA, an uncertainty-aware framework, for video anomaly detection (VAD) using vision-language models (VLMs). Experiments on UCF-Crime and XD-Violence datasets show UNA achieves superior scores (Tables 1–2) without instruction tuning, surpassing existing baselines.

**Strengths:**

- Originality: The paper proposes an uncertainty-aware view of VLM-based VAD, addressing reasoning-level uncertainty missed by earlier work.

- Quality: The proposed uncertainty tests based on temporal and semantic relevance (Eqs. 2 & 5; Fig. 3) are well-formulated and empirically supported (Table 3). The cooperative multi-agent strategy provides interpretable, improved predictions validated by ablation (Table 4). UNA achieves the best scores on both UCF-Crime and XD-Violence datasets. The paper provides case studies (Fig. 9) showing enhanced interpretability in ambiguous scenes.

- Clarity: Connections to uncertainty-aware LLMs and collective intelligence are clearly articulated.

- Significance: The paper introduces the uncertainty-aware reasoning mechanism for VLM-based VAD. It fills a research gap and could influence future multimodal explainability research.

**Weaknesses:**

- The probabilistic justification (Appendix A.2, Eqs. 7–10) is intuitive but lacks formal quantification of uncertainty propagation.
- Computational overhead and latency are mentioned qualitatively but not benchmarked against baselines.
- The contribution of individual hyperparameters is not reported beyond Appendix reference. And there is no sensitivity analysis provided.

**Questions:**

- Can authors provide with comparisons of computational efficiency (e.g., inference FPS vs. baseline VLMs)?
- Can authors clarify computational resource usage and memory scaling for multi-VLM cooperation?
- Can authors incorporate uncertainty calibration metrics to complement AUC results?
- Can authors conduct sensitivity studies on key hyperparameters to assess robustness?

---

### Official Review · Reviewer_4wZ6 · 2025-11-01

**Soundness:** 3
**Presentation:** 3
**Contribution:** 2
**Rating:** 2
**Confidence:** 4

**Summary:**

This paper proposes Uncertainty-Aware Vision-Language Framework (UNA) for Video Anomaly Detection, addressing a key limitation of existing vision-language models: their reasoning-level uncertainty when interpreting ambiguous or context-limited scenes. UNA identifies uncertainty by measuring prediction consistency across temporally and semantically relevant scenes, and mitigates it through collective intelligence, where multiple VLM agents collaborate to refine decisions. Without any fine-tuning, UNA achieves state-of-the-art performance on major VAD benchmarks such as UCF-Crime and XD-Violence, improving both accuracy and explainability. The framework demonstrates a general and scalable way to enhance VLM reasoning reliability, offering insights applicable to broader video understanding tasks.

**Strengths:**

1. This paper introduces an uncertainty-aware framework (UNA) that explicitly models and mitigates reasoning-level uncertainty in vision-language models.
2. The proposed method leverages temporal and semantic consistency for objective uncertainty detection.
3. The authors proposes a collective intelligence mechanism that enables multi-agent cooperation for more reliable reasoning
4. The proposed method achieves state-of-the-art results without fine-tuning, demonstrating strong generalization; and it enhances both the accuracy and interpretability of video anomaly detection in a principled and scalable way.

**Weaknesses:**

1. The paper’s core technical novelty is limited—framing uncertainty mitigation as "add a teammate" and ensembling multiple VLMs echoes a growing body of multi-agent and ensemble work, so the framework feels more incremental than foundational.
2. For the experimental section, the unexplained large drop of Holmes-VAD from 88.96 to 84.61 reported by the authors undermines trust in the evaluation protocol, and the paper omits several relevant baselines such as AnyAnomaly, EventVAD, and SUVAD that would be expected in a thorough comparison.
3. On XD-Violence the authors report AUC while average precision is the commonly used metric there, which complicates fair comparison and potentially overstates gains; missing metric-aligned baselines weakens the empirical claim.
4. The theoretical analysis given by Eq. 8 assumes conditional independence between two agents’ evidence given the label, but in practice the two VLMs share pretraining data, architectures and often prompts—violating that independence assumption and making the Bayesian argument overly simplistic.

**Questions:**

Due to the following concerns, I currently hold a negative opinion of this paper; however, I am willing to raise my score if the authors can provide clear and convincing clarifications.
1. The UNA framework largely builds on existing multi-agent or ensemble-style reasoning paradigms. Could the authors clarify what is conceptually novel about UNA compared with prior works that also mitigate VLM uncertainty through model cooperation or self-consistency? What unique insight or mechanism does UNA introduce beyond combining two agents?
2. The reported results for Holmes-VAD are substantially lower than its publicly verified benchmark performance (84.61 vs. 88.96), yet no explanation is given. Could the authors clarify the evaluation protocol, dataset splits, or preprocessing that led to this discrepancy? Is there any re-implementation detail that may have affected the numbers?
3. Several highly relevant methods such as AnyAnomaly, EventVAD, and SUVAD are not included in the comparison, and the authors report AUC instead of AP on XD-Violence, which differs from common practice. Could the authors justify these omissions and provide results under the standard evaluation metric to ensure fair comparison?
4. The framework assumes conditional independence between the two VLMs given the ground-truth label in Eq. 8. However, since both agents share pretraining data and often similar architectures, this assumption may not hold. How do the authors justify this assumption, and how sensitive is the framework’s effectiveness to correlated errors between agents?

---

### Note · Authors · 2025-11-12

I have read and agree with the venue's withdrawal policy on behalf of myself and my co-authors.